# Rapid Molecular Diagnostic Sensor Based on Ball-Lensed Optical Fibers

**DOI:** 10.3390/bios11040125

**Published:** 2021-04-15

**Authors:** Byungjun Park, Bonhan Koo, Jisub Kim, Kiri Lee, Hyeonjin Bang, Sung-Han Kim, Kyung Young Jhang, Yong Shin, Seungrag Lee

**Affiliations:** 1Medical Device Development Center, Osong Medical Innovation Foundation, 123 Osongsaengmyung-ro, Heungdeok-gu, Cheongju-si 28160, Korea; yachon@kbiohealth.kr (B.P.); jiseob@intekmedi.co.kr (J.K.); krlee@kbiohealth.kr (K.L.); crisenc@kbiohealth.kr (H.B.); 2School of Mechanical Engineering, Hanyang University, 222 Wangsimni-ro, Seongdong-gu, Seoul 04763, Korea; 3Department of Convergence Medicine, Asan Medical Institute of Convergence Science and Technology, Asan Medical Center, University of Ulsan College of Medicine, 88 Olympicro-43gil, Songpa-gu, Seoul 05505, Korea; qhsgksdl@ulsan.ac.kr; 4Department of Infectious Diseases, Asan Medical Center, University of Ulsan College of Medicine, 88 Olympicro-43gil, Songpa-gu, Seoul 05505, Korea; shkimmd@amc.seoul.kr; 5Department of Biotechnology, College of Life Science and Biotechnology, Yonsei University, 50 Yonsei Ro, Seodaemun-gu, Seoul 03722, Korea

**Keywords:** optical signal sensing, fiber optics, emerging infectious pathogens, rapid diagnosis, Q fever

## Abstract

Given the fatal health conditions caused by emerging infectious pathogens, such as severe acute respiratory syndrome coronavirus 2, their rapid diagnosis is required for preventing secondary infections and guiding correct treatments. Although various molecular diagnostic methods based on nucleic acid amplification have been suggested as gold standards for identifying different species, these methods are not suitable for the rapid diagnosis of pathogens owing to their long result acquisition times and complexity. In this study, we developed a rapid bio-optical sensor that uses a ball-lensed optical fiber (BLOF) probe and an automatic analysis platform to precisely diagnose infectious pathogens. The BLOF probe is easy to align and has a high optical sensing sensitivity (1.5-fold) and a large detection range (1.2-fold) for an automatic optical sensing system. Automatic signal processing of up to 250 copies/reaction of DNA of Q-fever-causing *Coxiella burnetii* was achieved within 8 min. The clinical utility of this system was demonstrated with 18 clinical specimens (9 Q-fever and 9 other febrile disease samples) by measuring the resonant wavelength shift of positive or negative samples for *Coxiella burnetii* DNA. The results from the system revealed the stable and automatic optical signal measurement of DNA with 100% accuracy. We envision that this BLOF probe-based sensor would be a practical tool for the rapid, simple, and sensitive diagnosis of emerging infectious pathogens.

## 1. Introduction

The social and environmental changes caused by globalization and climate change have resulted in the wide spread occurrence of newly emerging and re-emerging zoonotic infectious agents [1,2,3]. In particular, emerging infectious diseases with pandemic potential, such as severe acute respiratory syndrome (SARS), coronavirus disease 2019 (COVID-19), malaria, Ebola virus disease, Middle East respiratory syndrome (MERS), and Zika virus disease, pose significant risks to global societies and public health [4,5,6]. With the current advances in the field of medical diagnostics, rapid and accurate techniques based on nucleic acid detection approaches have become available for detecting emerging infectious pathogens [7,8]. Among such approaches, the polymerase chain reaction (PCR) is considered a gold standard in the diagnostic field. However, conventional PCR methods require a long processing time, high costs, and technical expertise and have demonstrated low sensitivity in clinical use and limitations in point of care testing. Although efforts to overcome these PCR problems are continuing, the development of a rapid and accurate method for detecting highly infectious pathogens remains a major challenge [9].

Recently, optical sensors have been widely used for the detection and analysis of proteins and genomes as well as for the diagnosis of infectious diseases [10]. The techniques rely on light absorbance, fluorescence, luminescence, Raman scattering, or refractive indexes and are rapid, real-time, robust, and more sensitive technologies than electrochemical and mechanical ones [10,11,12]. In particular, because optical fibers enable remote sensing, are cost effective, and have high platform flexibility, they have been used in biosensors to detect changes in optical signals and to observe various biological phenomena [13,14,15,16]. Because of these advantages, myriad techniques based on optical sensors have been developed for the detection of biomolecules in the field of disease diagnostics [17,18,19].

We have previously proposed a bio-optical sensor for the rapid diagnosis and identification of emerging infectious pathogens [20,21,22,23,24]. The sensor could provide for the highly accurate and rapid (i.e., within 20 min) diagnosis and identification of several emerging pathogens, such as Middle East respiratory syndrome coronavirus (MERS-CoV), human coronavirus (HCoV), Ebola virus, Zika virus, and *Coxiella burnetii* (the causative agent of Q fever) [21,22,23,24]. However, the use of this bio-optical sensor for the detection of biomolecules in various applications still had limitations, such as the complexity in the operation of the system, the difficulty in aligning the optical fiber to the waveguide, and data analysis can be conducted manually. Therefore, to allow use of the bio-optical sensor for disease diagnostics in real-world settings, a novel optical sensing platform that can overcome the above-mentioned limitations is needed.

Herein, we report the development of a novel molecular diagnostic system for the rapid detection of pathogens. This bio-optical sensing system uses a ball-lensed optical fiber (BLOF) probe on a silicon microring resonator (SMR) sensor to enable stable measurements and accurate diagnoses. Similar to an objective lens, the BLOF generates a depth of focus (DOF) of a laser beam, resulting in excitation of the focused beam toward the SMR sensor, which provides more stable measurements than those obtained with our previous bio-optical sensor and at a higher efficiency. The input channel of the single chamber is located within the distance of the DOF along the optical axis without the need to place the fibers near the SMR sensor. To define the optimal distance and the range of the optical signal measurement between the BLOF and the SMR sensor, we simulated the optical properties of the lens under the same conditions as those of currently available ball lensed fibers. By comparing the optical signal intensity of the BLOFs with that of the previous optical fiber, we showed that the stability of the BLOF-based sensor allows for a larger measurement range. Furthermore, we validated the clinical utility of this system with 18 specimens (9 from patients with Q fever and 9 from those with other febrile diseases) within 8 min in a label-free and real-time manner and automatically analyzed and processed the resonant wavelength data. Accordingly, this BLOF-based molecular diagnostic technique using an automatic optical sensing system is useful for the rapid, user-friendly, and highly sensitive diagnosis of biomolecules in the clinical setting.

## 2. Materials and Methods

### 2.1. BLOF Based Optical Bio-Signal Measurement System

The schematic design and image of our BLOF-based bio-optical signal measurement system is shown in Figure 1. The basic configuration of this system comprises an illumination part with a swept source laser as the optical source (Model TSL-550; Santec, Komaki, Japan) and a detection part with a photodiode as the optical signal acquisition device. The central wavelength of the swept source laser is 1550 nm and its scan range is 50 nm. The workflow of the device is as follows: a swept laser beam with a 10 pm sweeping step passes through a BLOF and enters the single chamber of the bio-optical sensor on the sample stage. The bio-optical sensor then amplifies and detects the target DNA at a given laser wavelength. The optical signal related to the target DNA is then gathered into a BLOF connected to a photodiode (Model C10439-8271; Hamamatsu Photonics, Shizuoka, Japan), which detects the signal. A data acquisition system (DAQ, National Instrument, Seoul, Korea) was used for the signal processing and automatic system control in a personal computer. Two 3-axis motor-controlled stages were used to automatically manipulate the position of the BLOF toward the bio-optical sensor. A polarization-maintaining optical fiber (PMF) was used to manufacture the BLOF.

### 2.2. Signal Detection with the BLOF-Based Bio-Optical Sensor

To detect optical signals using a SMR sensor, we manufactured the BLOF using a tungsten filament fusing splicer device (Model GPX-3000; Thorlabs, New Jersey, USA). The BLOF was composed of three parts with the PMF, a core less fiber (CLF), and a ball lens. The PMF had a core diameter of 8.5 μm and a cladding diameter of 125 μm. The CLF had a cladding diameter of 125 um. First of all, the polished PMF was spliced to the polished CLF with a specific length using a fusing splicer device [25]. Secondly, we followed the fabrication process of a ball lens, mentioned by the previous paper [25,26]. Finally, the distal end of the CLF was heated to form a ball lens with the specific diameter (≅300 um) and the spherical refraction surface by controlling the temperature of the filament and the heating time on the splicer device. The fabricated BLOF had the spherical refraction surface with the radius of curvature (≅0.15 mm) and provided the function of convergence lens with a focal length (≅1.5 mm).

The positions of two BLOFs were automatically adjusted toward the input and output ports of the SMR sensor, respectively, by finding the maximum point of the optical signal measured from the photodiode (Figure 1). After completion of the BLOF adjustment process, a swept laser beam was focused onto the SMR sensor through the input BLOF, and the laser beam emitted from the SMR sensor was gathered into the output BLOF connected to the photodiode. Finally, the optical signal sensitive to the specific wavelength of the swept laser beam was detected in the bio-optical sensor.

### 2.3. Automatic Analysis of Detection of the Resonant Wavelength Shift in a Given Time

Detection of the resonant wavelength shift was carried out using the method reported in our previous studies [20,21,22,23,24]. In this study, we included Gaussian smoothing in the signal processing process to filter out any signal noise in the detected resonant wavelength information. We also applied weighted mean processing to determinate the weight position of the resonant wavelength range modified by the signal processing, defined as
(1)$$\mathrm{Weighted}\;\mathrm{Mean}=\frac{\sum_{i=1}^{n}\left(x_{i}\times w_{i}\right)}{\sum_{i=1}^{n}w_{i}}$$
where $x_{i}$ is the value and $w_{i}$ is the weight. Using these two signal processing techniques, we acquired all the resonant wavelength information obtained by the SMR sensor in a time cycle and used it to calculate the change in resonant wavelength shift. Figure 2 illustrates the steps taken to set up the automatic optical sensing measurement system, including how to measure the change in the resonant wavelength in a given time and to operate the automatic system using our developed software program. The first step was the optical alignment of the BLOF through monitoring of the optical signal intensity. Figure 2A illustrates the process used to find the optimal position of the two optical fiber probes in the input and output ports. After finding the maximum position within the measured optical signals (the white arrow in Figure 2A), the BLOF was moved to the maximum position (the red circle in Figure 2A). The second step (Figure 2B) involved biosensor measurement of the shifted resonant wavelength in a given time. As shown in Figure 2B, after acquiring the resonant wavelengths over 20 min, their peak positions were detected after weighted mean processing (the white dashed box of ①) and the shifted resonant wavelengths were calculated in relation to the wavelength at 0 min (the white dashed box of ②).

### 2.4. Amplification of Coxiella Burnetii DNA on the BLOF Based Bio-Optical Sensor

To adapt the BLOF-based bio-optical sensor for the rapid diagnosis of pathogen DNA, the SMR was used as the sensor device, as previously described [20,23,24]. To use the SMR sensor for detecting *C. burnetii* DNA, the SMR sensor was first immersed for 2 h in a silanization solution made up of 2% 3-aminopropyltriethoxysilane in an ethanol/H_2_O mixture (95%/5%, *v*/*v*) and then thoroughly rinsed with ethanol and deionized (DI) water. Then, the SMR sensor was incubated for 1 h with 2.5% glutaraldehyde in DI water containing 5 mM sodium cyanoborohydride, then rinsed with DI water, and finally dried under N_2_ gas. For immobilization of the solid-phase forward primer, the pretreated SMR sensor was incubated overnight at ambient temperature with a 1 mM primer solution containing 5 mM sodium cyanoborohydride, then rinsed with DI water, and finally dried under N_2_ gas. For amplification of the pathogen DNA on the SMR sensor surface, the isothermal recombinase polymerase amplification assay was used as previously described [20,23,24].

## 3. Results

### 3.1. Design and Manufacture of the Ball-Lensed Optical Fibers (BLOFs)

To manufacture BLOFs for our bio-optical sensor system, we simulated the same optical properties as those of currently available ball lensed fibers using Zemax software (shown in Appendix A) and designed the optical sensing probe to have a lens like that of the available ball lensed fibers. Figure 3A shows the structure of the BLOF, which consists of a PMF, CLF, and a ball lens. The CLF, which had a refractive index of 1.444 at 1550 nm, was designed to have a diameter of 285 μm, and the ball lens had a diameter of 300 μm. Figure 3B shows the BLOF simulation results. A virtual laser beam with a wavelength of 1550 nm was aimed into the PMF, and the emitted beam converged after passing through the CLF and the ball lens. Figure 4A shows the intensity profile of the beam at the focal plane (*z* = 1.5 mm). The distance from the BLOF to the focal plane, known as the focal point, was 1.5 mm. Figure 4B shows the intensity line profile along the horizontal arrow in Figure 4A, from which the full width at half maximum (FWHM) of the focused beam was calculated to be 24.7 μm, revealing that the BLOF could act as a converging lens to precisely focus the beam into the waveguide of the SMR sensor. Figure 4C shows the dependence of the FWHM of the beam on the distance of the laser beam emitted from the BLOF. Unlike a regular optical fiber, the BLOF creates a convergent beam width at 1.5 mm. From this finding, the BLOF was considered suitable for our proposed optical signal sensing system because it could transfer a larger amount of the signal intensity to the SMR sensor than our previous optical fiber could, thereby vastly improving the efficiency of the optical signal sensor. We also confirmed that the beam width did not differ significantly within ±200 μm from the focal point of the laser beam, which meant that the BLOF provided a larger optical signal sensing range and was thus a more stable measurement system than our previous system without the ball lens.

As mentioned above, the BLOFs had been manufactured after taking simulated lens conditions into consideration (Figure 3A). Figure 4D shows the manufacture of the BLOF using a fusion splicing machine, which fabricates the junctions and ball lenses in optical fibers. The distance of the BLOF along the vertical axis was approximately 300 μm. We assumed the BLOF had the same spherical refraction surface and radius of curvature as the simulated lens in our study since the curvature of the fabricated BLOF (Curvature ≅ 6.67 mm^−1^) correlated well with that of the simulated lens. We confirmed the formation of the ball lens at the end of the CLF, which functions to control the degree of focusing depending on the radius of curvature and refractive index of the lens material. To demonstrate the utility of our proposed BLOF-based bio-optical sensing system, we measured the optical signal intensities along the distance of the BLOF from the bio-optical sensor and compared the results with those of our previous optical fiber under the same detection conditions. As shown in Figure 1, the two BLOF-based optical signal sensing probes (OSSPs) were tilted at a slight angle of approximately 8° with respect to the *y* direction. We fixed the position of the OSSP that was connected to the photodiode and manipulated the position of the other OSSP along the tilt angle direction at the input end of the bio-optical sensor.

### 3.2. Characterization of the Ball-Lensed Optical Fibers (BLOFs)

To eliminate the irregular refraction of a beam due to a rough optical fiber surface, an optical fiber with its end evenly polished was also tested using this proposed system (Figure 1) for comparison with the BLOF. Figure 5 shows the results of the optical signal intensities detected using the BLOF and the polished optical fiber, where the *x* axis represents the relative distances for the OSSP with the BLOF and the OSSP with the polished optical fiber. First, while monitoring the maximum value of the optical signal, we moved the polished optical fiber closer to the bio-optical sensor to excite the laser beam into the sensor input channel. The distance between the polished optical fiber and the input sensor was approximately 0.05 mm for the maximum optical signal intensity. We moved the polished optical fiber from the point of maximum intensity to 0.3 mm with a step size of 0.05 mm. The intensity decreased significantly when the distance was 0.1 mm away from the starting point. On the other hand, the maximum signal intensity of the BLOF was measured at approximately 1.5 mm from the starting intensity point of the polished optical fiber, which meant that the BLOF functioned as a lens with a focal length of approximately 1.5 mm. We moved the BLOF from 0.95 mm to 2.15 mm with a step size of 0.050 mm. From these results, it was found that the optical sensing range covered by the BLOF was 1.2-fold wider than that of the polished optical fiber, and the detected intensity at the maximum intensity point was approximately 1.5-fold higher. These results demonstrate the improved stability and sensitivity of the new BLOF-based OSSP proposed in Figure 1, given the method provided the system with the capability to cover a larger range of measurements with a higher signal intensity. Our findings are of great importance in terms of the clinical application of the system for the rapid diagnosis of infectious pathogens [20,21,22,23,24]. In our previous system, the optical fiber could easily come into contact with the bio-optical sensor, even with only small external impact, because the distance between the two components was less than 0.1 mm, which caused misalignment of the measurement angle by 8° and damage to the sensing part of the system, such as fractures. By contrast, the new system obviates contact problems, as the distance between the BLOF and the sensor is much wider. Therefore, the BLOF based bio-optical sensing system provides automatic operation, measurement and analysis for the detection of pathogens.

### 3.3. Clinical Diagnosis of Q Fever Using the BLOF Based Sensor

To optimize and validate our new BLOF-based bio-optical sensor for clinical applications, we used it to measure the quantitative wavelength shifts under *C. burnetii* DNA-positive and -negative conditions and to analyze clinical samples from nine patients with Q fever and nine patients with other febrile diseases (Figure 6 and Appendix A). The forward and reverse primers for the BLOF-based bio-optical sensor were synthesized at the usual length of approximately 34 bp (Appendix A). To compare the differences in wavelength shifts of the amplified *C. burnetii* DNA, we measured the quantitative wavelength shifts under DNA-positive (using different DNA concentrations; i.e., 2.5 × 10^4^, 2.5 × 10^3^ and 2.5 × 10^2^ copies/reaction) and DNA-negative conditions every 5 min for 30 min. Figure 6A shows the temporal changes in the resonant wavelength shifts (∆pm) for a given concentration of DNA copies/reaction and under negative conditions, where a vertical value represents the resonant wavelength shift (in pm) and a horizontal value represents the time. It was confirmed that the amount of wavelength shifts increased overall with increasing DNA concentration and time. The wavelength shift of *C. burnetii* DNA as the positive control was higher than that of distilled water as the negative control. Next, the linear relationship between the wavelength shift and the target concentration over 25 min was obtained (Figure 6B). From these results, we confirmed that the results obtained with the proposed system agreed well with those obtained with the previous system described [23,24]. Moreover, the proposed system would be able to recognize subtle changes in the pathogen concentration from earlier stages of the disease within 20 min, as reported in our previous studies [20,21,22,23,24]. In this respect, we expect that the proposed method will provide the rapid diagnosis of infectious pathogens with high sensitivity and stability.

Finally, we proved that our system could be a practical medical diagnostic device by measuring the resonant wavelength shifts in blood samples from nine patients with Q fever and nine patients with other febrile diseases under *C. burnetii* DNA-positive and -negative conditions. The 18 blood plasma samples were obtained using protocols approved by the Institutional Review Board of Asan Medical Center (2018–9023) with informed consent from the patients, Republic of Korea [24,27]. DNAs were extracted from 200 μL of the blood plasma sample using the QIAamp DNA Mini Kit (Qiagen, Hilden, Germany) and eluted with approximately 100 μL of elution buffer. Conventional PCR was performed using forward and reverse primers, which were designed to detect the *C. burnetii* gene coding for IS1111a transposase elements, to amplify DNA and detect *C. burnetii* for confirmation of Q fever patients (Appendix A). Figure 6C shows the bar chart of the ∆pm over 8 min per sample under positive and negative conditions, where a vertical value represents the resonant wavelength (in pm) and the error bar indicates the standard deviation of the mean. The BLOF-based bio-optical sensor accurately diagnosed the presence of *C. burnetii* DNA in patients with Q fever and its absence in other febrile diseases, with a sensitivity and specificity of 100%. Taking these results together, it was concluded that the BLOF-based bio-optical sensor could rapidly and accurately detect *C. burnetii* DNA and diagnose Q fever, allowing us to distinguish patients with Q fever from those with other febrile diseases with minimum handling.

## 4. Conclusions

In summary, we have developed a novel molecular diagnostic device consisting of a BLOF-based optical signal measurement system and an SMR-based bio-optical sensor for the determination of pathogens in a rapid and automatic manner. We have improved the stability and sensitivity of our previous bio-optical sensing system by integrating it with the BLOF technique as an optical sensing probe. The new system can provide a 1.2-fold wider optical sensing range with 1.5-fold higher signal intensity. The further advantages of this system are its rapidity, simplicity, isothermal nucleic acid amplification capability (38 °C), and label-free diagnostics, allowing a new multidisciplinary approach for the diagnosis of emerging infectious diseases. We confirmed that the limit of detection of this novel system was approximately 2.5 × 10^2^ copies/reaction. Additionally, the applicability of the system in a clinical setting was demonstrated by confirming the diagnosis of nine patients with Q fever and nine patients with other febrile diseases with high sensitivity (100%) and specificity (100%). We anticipate that our novel BLOF-based bio-optical sensing system will be effectively used as one of the standard methods for the rapid diagnosis of the causative pathogens of emerging infectious diseases, such as SARS, COVID-19, MERS, HCoV disease, Zika disease, Ebola, and Q fever.

## Figures and Tables

**Figure 1 biosensors-11-00125-f001:**
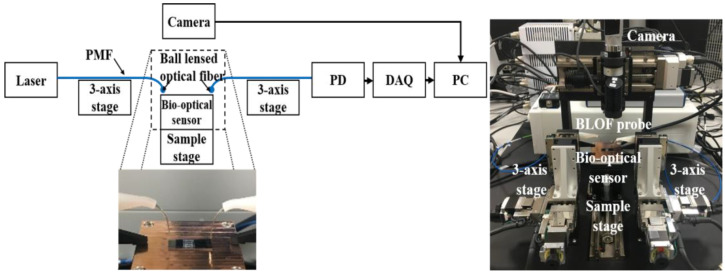
Schematic diagram and photograph of the BLOF based bio-optical signal sensing and measurement system.

**Figure 2 biosensors-11-00125-f002:**
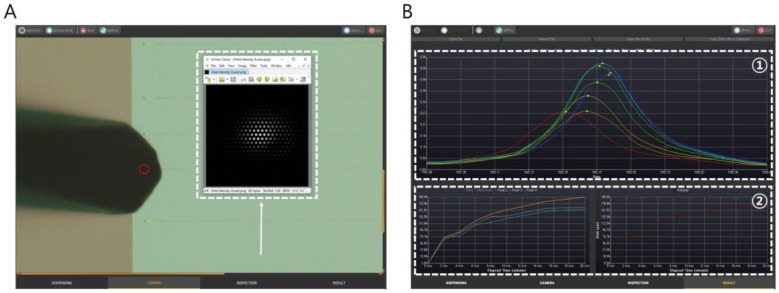
Illustration of the automatic optical sensing measurement system using our developed software program. (**A**) Optical alignment of the ball-lensed optical fiber (BLOF) through monitoring of the optical signal intensity. (**B**) Biosensor measurement of the shift in resonant wavelengths in a given time (0–20 min).

**Figure 3 biosensors-11-00125-f003:**
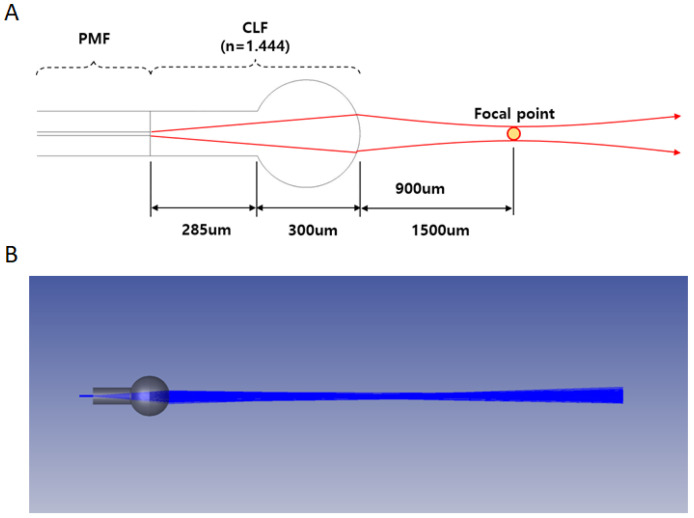
Schematic representation of the ball-lensed optical fiber (BLOF). (**A**) Structure of the BLOF. (**B**) Simulated results of the BLOF using Zemax software.

**Figure 4 biosensors-11-00125-f004:**
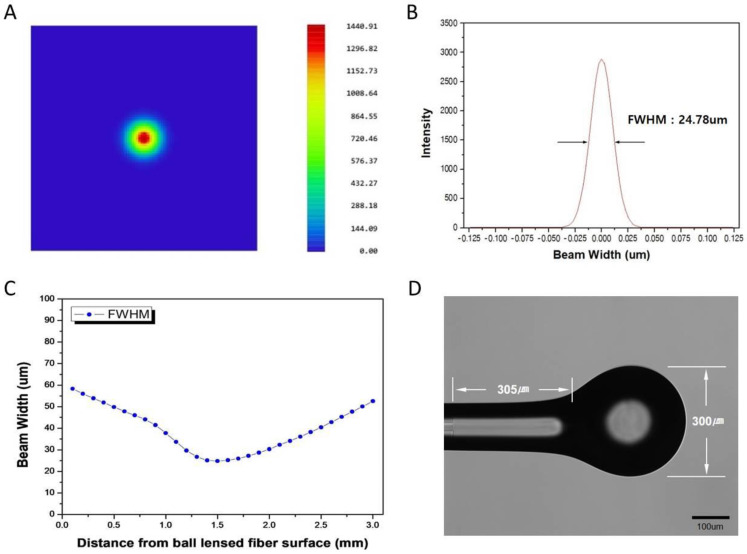
Information on the ball-lensed optical fiber (BLOF). (**A**) Intensity profile of a beam at the focal plane (*z* = 1.5 mm). (**B**) Intensity line profile along the horizontal arrow. (**C**) Beam widths according to the distances of the laser beam emitted from the BLOF. (**D**) Manufactured BLOF using a fusion splicing machine.

**Figure 5 biosensors-11-00125-f005:**
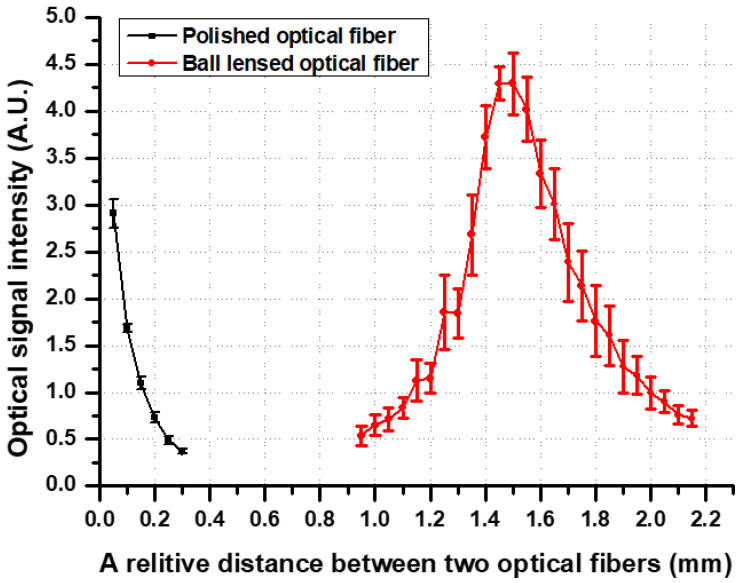
Results of the detected optical signal intensities using the ball-lensed optical fiber (BLOF) and the polished optical fiber.

**Figure 6 biosensors-11-00125-f006:**
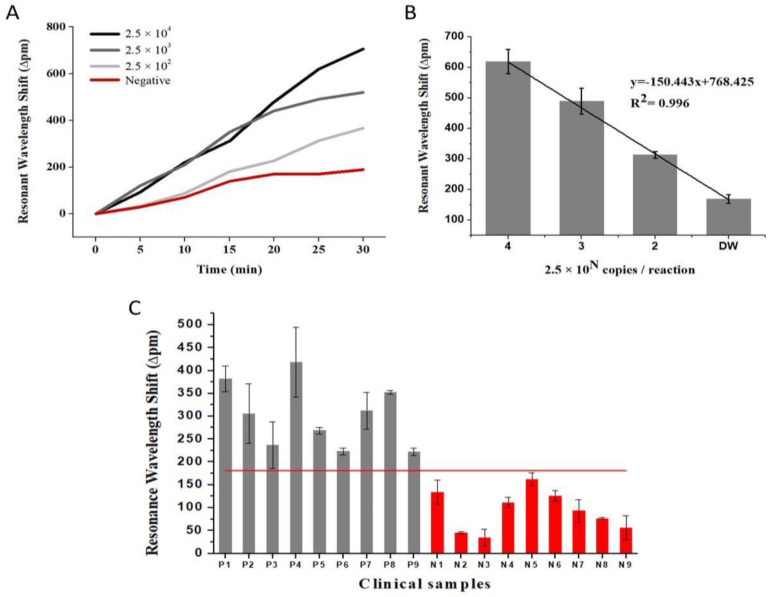
Optimization and validation of the ball-lensed optical fiber (BLOF) bio-optical sensor for clinical application, using *Coxiella burnetii* DNA and clinical samples. (**A**) Limit of detection of the bio-optical sensor assay for amplified *C. burnetii* DNA. The colors represent the amount of the target: 2.5 × 10^4^ copies/reaction (black); 2.5 × 10^3^ copies/reaction (gray with darker 75%); 2.5 × 10^2^ copies/reaction (gray with darker 50%); and negative control (red). (**B**) Linear relationship between the wavelength shift and the target concentration in 25 min. (**C**) Bar chart showing the resonant wavelength shift results in 8 min per sample. The colors represent the detection of *C. burnetii* DNA (P, gray) in 9 blood samples from patients with Q fever and of no *C. burnetii* DNA (N, red) in 9 blood samples from patients with other febrile diseases. The red line indicates a cut off for positive and negative in 8 min. Error bars indicate the standard deviation of the mean.

## Data Availability

Not applicable.

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
