# Peer review of "Rapid Molecular Diagnostic Sensor Based on Ball-Lensed Optical Fibers"

_biosensors, 2021, doi:10.3390/bios11040125_

Round 1
Reviewer 1 Report
The main body of this article is the development of a ball-lensed optical fiber (BLOF) probe and the automatic analysis platform in the detecting system based on a silicon microcircle resonator. Considering their previous works about the same research, some new progresses are achieved in this article. So I think it is acceptable for the journal of Biosensors.
Besides, I have some suggestion for the authors,
- The structure of the article could be rearranged and modified better: the contents about the BLOF, such as the design and manufacture, the alignment to the optical sensor, the characters, are supposed to be reorganized together in the angle of optics. Without interrupted by the sample treatments. The part of 2.5 could be moved into part 3.3.
- The relationship of the size and shape of BLOF to the sensitivity and efficiency of the bio-optical sensing system are not mentioned in this research. How could you ensure to fabricate the BLOF in shape and size exactly the same with the design and optimization results?
Author Response
Reviewer #1: The main body of this article is the development of a ball-lensed optical fiber (BLOF) probe and the automatic analysis platform in the detecting system based on a silicon microcircle resonator. Considering their previous works about the same research, some new progresses are achieved in this article. So I think it is acceptable for the journal of Biosensors.
- We thank the reviewer for very careful reading of the manuscript and detailed suggestions that have helped us improve significantly the quality of the manuscript.
Besides, I have some suggestion for the authors,
- The structure of the article could be rearranged and modified better: the contents about the BLOF, such as the design and manufacture, the alignment to the optical sensor, the characters, are supposed to be reorganized together in the angle of optics. Without interrupted by the sample treatments. The part of 2.5 could be moved into part 3.3.
- We thank the reviewer for the comment. According to the reviewer’s comment, we have moved the “Figure 3 and 4” into “3.1. Design and manufacture of the ball-lensed optical fibers (BLOFs)” in the revised manuscript.
- We have also moved the “Figure 5” into “3.2. Characterization of the ball-lensed optical fibers (BLOFs)” and have moved “Figure 6” into “3.3. Clinical Diagnosis of Q fever using the BLOF based sensor” in the revised manuscript.
- According to the reviewer’s comment, the part 2.5 has been moved to part 3.3 in the revised manuscript.
On Page 7, “The forward and reverse primers for BLOF-based bio-optical sensor were synthesized at the usual length of approximately 34 bp (Table S1).”
On Page 8, “The 18 blood plasma samples were obtained using protocols approved by the Institutional Review Board of Asan Medical Center (2018–9023) with informed consent from the patients, Republic of Korea [24, 27]. DNAs were extracted from 200 μL of the blood plasma sample using the QIAamp DNA Mini Kit (Qiagen, Hilden, Germany) and eluted with approximately 100 μL of elution buffer. Conventional PCR was performed using forward and reverse primers, which were designed to detect the C. burnetii gene coding for IS1111a transposase elements, to amplify DNA and detect C. burnetii for confirmation of Q fever patients (Figure S1 and Table S1).”
- The relationship of the size and shape of BLOF to the sensitivity and efficiency of the bio-optical sensing system are not mentioned in this research. How could you ensure to fabricate the BLOF in shape and size exactly the same with the design and optimization results?
- We thank the reviewer for the comment. We firstly determined the distance of 1.5 mm between BLOF and the SMR sensor to provide more stable optical sensing measurement than our previous measurement, which was appropriate to automatically align BLOF probe to the SMR sensor without any impact. We designed BLOF with the five different sizes and calculated the beam width for 1.5 mm distance between BLOF and the SMR sensor. Here, the curvature of BLOF has value, which correlated well with the simulated lens. Following figure showed BLOF with a size of 300 um had the minimum beam width of a laser beam at 1.5 mm distance, which meant the laser beam through BLOF with a size of 300 um was most transmitted to the SMR sensor.
In the specific optical design of BLOF, we optimized the fabrication process of BLOF with a size of approximately 300um and a spherical refraction surface. We explained the fabrication process of BLOF in the comment of Reviewer #3. Figure 5 showed the fabricated BLOF had the maximum optical signal intensity at approximately 1.5mm distance between BLOF and the SMR sensor and 1.5-fold higher optical signal intensity than the previous fiber. It was possible to mention the efficiency and a sensitivity of the optical sensing system in the specific size and the spherical refraction surface of BLOF in our study, but it was hard to be satisfied with them in BLOF with a different size since the size of BLOF was different according to the distance between BLOF and the SMR sensor.
On Page 6, “We assumed BLOF had the same spherical refraction surface and radius of curvature as the simulated lens in our study since the curvature (c) of the fabricated BLOF (c 6.67 mm-1) correlated well with that of the simulated lens.”

Reviewer 2 Report
- The limit of detection of the proposed sensor should to be calculated using the formulations given in the literature.
- The number of samples with two different concentrations to generate the calibration curve is not satisfactory. Minimum four sample concentrations should be used.
- The study is using clinical samples to evaluate the proposed sensor however the control samples used in the study is only water. To represent a clinical utility the study should include negative plasma samples as control samples.
Author Response
Reviewer #2: The limit of detection of the proposed sensor should to be calculated using the formulations given in the literature.
- We thank the reviewer for very careful reading of the manuscript and detailed suggestions that have helped us improve significantly the quality of the manuscript.
- We have measured the resonant wavelength shift using serially diluted DNAs and RNAs to confirm the detection limit of SMR sensor as previously described [20-24]. In this study, we used the serial diluted burnetii DNAs ( - copies / reaction) to generate the calibration curve and confirm detection limit of proposed sensor. In addition, all results of the resonant wavelength shift obtained through the BLOF sensor were automatically analyzed and calculated using the formulation ( ) given in the manuscript.
The number of samples with two different concentrations to generate the calibration curve is not satisfactory. Minimum four sample concentrations should be used.
- We thank the reviewer for the comment. We have revised the manuscript and added results of resonant wavelength shift using copies / reaction to generate the calibration curve in revised Figure 6A-B. We tried to confirm the detection limit of the proposed sensor using samples of - copies / reaction, and finally confirmed that the detection limit was copies / reaction.
Figure 6A-B
The study is using clinical samples to evaluate the proposed sensor however the control samples used in the study is only water. To represent a clinical utility the study should include negative plasma samples as control samples.
- We thank the reviewer for the comment. We analyzed DNA from 18 clinical plasma samples, including those of 9 burnetii infected patients (DNA positive) and 9 other febrile diseases patients (DNA negative). This result is shown in figure 6C. As a result, we used DW and 9 plasma samples of other febrile diseases patients as negative control to evaluate the proposed sensor.
Figure 6C

Reviewer 3 Report
In this manuscript the authors describe a new method for DNA detection based diagnostic. While this is a new interesting method, it is far from other comparable methods of DNA detection. However, I would recommend for publication due to the novelty of detection method for its potential to be interesting to selective the readers. Here are my recommendations for betterment of the manuscript.
- Provide some more details of the manufacturing controls and charecterization of the BLOF.
- How the cutoff line was chosen for the +ve vs -ve samples in Fig 6c. It seems very random. The authors should consider an objective way of determining the cutoff line.
Author Response
Reviewer #3: In this manuscript the authors describe a new method for DNA detection based diagnostic. While this is a new interesting method, it is far from other comparable methods of DNA detection. However, I would recommend for publication due to the novelty of detection method for its potential to be interesting to selective the readers. Here are my recommendations for betterment of the manuscript.
- We thank the reviewer for very careful reading of the manuscript and detailed suggestions that have helped us improve significantly the quality of the manuscript.
- Provide some more details of the manufacturing controls and characterization of the BLOF.
- We thank the reviewer for the comment. We have revised the manuscript and added some more details of the manufacturing controls and characterization of the BLOF.
On Page 3, “To detect optical signals using a SMR sensor, we manufactured BLOF using a tungsten filament fusing splicer device (Model GPX-3000; Vytran). BLOF was composed of three parts with PMF, core less fiber (CLF), and a ball lens. PMF had a core diameter of 8.5 μm and a cladding diameter of 125 μm. CLF had a cladding diameter of 125 um. First of all, the polished PMF was spliced to the polished CLF with specific length using a fusing splicer device [25]. Secondly, we followed the fabrication process of a ball lens, mentioned by the previous paper [25, 26]. Finally, the distal end of CLF was heated to form a ball lens with the specific diameter ( 300 um) and the spherical refraction surface by controlling the temperature of the filament and the heating time on the splicer device. The fabricated BLOF had the spherical refraction surface with the radius of curvature ( 0.15 mm) and provided the function of convergence lens with a focal length ( 1.5 mm).”
We have added two extra reference papers were added to the reference list of the paper.
[25]. Park, B.J.; Lee, S.R.; Bang, H.J.; Kim, B.Y.; Park, J.H.; Kim, D.G.; Park, S.S.; Won, Y.J. Image-guided laparoscopic surgical tool (IGLaST) based on the optical frequency domain imaging (OFDI) to prevent bleeding. Sensors (Switzerland) 2017, 17, 919
[26]. K.M.; Shishkov, M.; Chee, A.; Applegate, M.B.; Bouma, B.E.; Suter, M.J. Flexible transbronchial optical frequency domain imaging smart needle for biopsy guidance. Biomed. Opt. Express 2012, 3, 1947-1954
- How the cutoff line was chosen for the +ve vs -ve samples in Fig 6c. It seems very random. The authors should consider an objective way of determining the cutoff line.
- We thank the reviewer for the comment. In Fig 6C, we have added red line to indicate a cut off for the determination of +ve and -ve samples within 8 min. The resonant wavelength of non-specific binding on the SMR sensor was shifted depends on the oligonucleotide sequences, the type of clinical sample, and the reaction time that was previously reported [20-24]. Based on our previous reports, we performed blind tests with 18 blood plasma samples and measured resonant wavelength shift up to 20 minutes to compare the resonant wavelengths between +ve and -ve samples. We established the detection criterion based on the resonant wavelength obtained from the non-specific binding.
Round 2
Reviewer 2 Report
I thank the authors for revising the manuscript.